# Modification of Apremilast from Pills to Aerosol a Future Concept

**DOI:** 10.3390/ijerph182111590

**Published:** 2021-11-04

**Authors:** Paul Zarogoulidis, Christoforos Kosmidis, Nikolaos Kougkas, Aimilios Lallas, Dimitris Petridis, Wolfgang Hohenforst-Schmidt, Haidong Huang, Lutz Freitag, Chrisanthi Sardeli

**Affiliations:** 1Pulmonary Department, General Clinic, Euromedica Private Hospital, 546 45 Thessaloniki, Greece; 23rd Surgery Department, AHEPA University General Hospital, Aristotle University of Thessaloniki, 546 21 Thessaloniki, Greece; dr.ckosmidis@gmail.com; 3Rheumatology Department, IPPOKRATEIO University General Hospital, Aristotle University of Thessaloniki, 541 24 Thessaloniki, Greece; nkougas@yahoo.gr; 41st Dermatology Department, Aristotle University of Thessaloniki, 540 06 Thessaloniki, Greece; emlallas@gmail.com; 5Department of Food Science and Technology, International Hellenic University, 507 01 Thessaloniki, Greece; petridis@food.teithe.gr; 6Sana Clinic Group Franken, Department of Cardiology, Pulmonology, Intensive Care, Nephrology, Hof Clinics, University of Erlangen, 91054 Hof, Germany; w.h-s@gmx.de; 7Department of Respiratory and Critical Care Medicine, First Affiliated Hospital of Naval Medical University (Changhai Hospital, Second Military Medical University), Shanghai 200433, China; hhdongbs@126.com; 8Department of Pulmonology, University Hospital Zurich, 8091 Zurich, Switzerland; Freitag-Hemer@t-online.de; 9Department of Pharmacology & Clinical Pharmacology, School of Medicine, Faculty of Health Sciences, Aristotle University of Thessaloniki, 546 45 Thessaloniki, Greece; sardeli@auth.gr

**Keywords:** nebulisers, jet-nebulisers, ultrasound nebulisers, apremilast, psoriasis, mastersizer

## Abstract

Background: Inhaled drugs have been available in the market for several years and for several diseases. Drugs for chronic obstructive pulmonary disease, cystic fibrosis, and diabetes have been used for several years. In the field of drug modification, these drugs range from tablets to aerosol. Methods: Milling as used to break down the tablets to powder and nebulisers are used to produce aerosol droplets. A mastersizer was used to measure the mass median aerodynamic diameter of the aerosol droplets. Results: Apremilast produced mmad diameters (2.43 μm) without any statistical difference between the different jet-nebulizers. The residual cup B contributed to greater mmad diameters as the 95% interval of mean values, based on those the ANOVA mean square clearly indicated, followed by cups C and F. The previous interval plot is much better clarified when the interaction means between drug and residual cap are plotted. The residual cups B, C and F produce mmad between (2.0–3.2). Conclusion: In the current research study we demonstrated our methodology to create apremilast powder and produce apremilast aerosol droplets with different nebulisers and residual cups.

## 1. Introduction

Inhaled medications have been available in the market for several years. The most common drugs administered through inhalation are the drugs used for chronic obstructive pulmonary disease (COPD) or bronchoconstriction [1,2]. Inhaled antibiotics have been also used and are in currently use for cystic fibrosis and copd patients [3]. Several efforts have been made to investigate different antibiotics as aerosol administration in the laboratory [4]. In the past decade insulin was admnistered as an aerosol. However, due to cost effectiveness difficiulties, its production was terminated [5]. Several efforts have been made since then to investigate different insulin products as aerosols in the laboratory [6]. Inhaled chemotherapy has been also investigated in the laboratory setting, and administered in clinical trials for several types of primary cancers and lung metastasis [7,8,9,10]. Inhaled gene therapy has been also investigated [11]. Moreover, there are several drugs are also being investigated to determine whether they could be administered as aerosols [4,6,7,12,13]. It is known that there are several factors affecting aerosol production and deposition within the airways. The major obstacles for efficient aerosol deposition are the defense mechanisms of the airways [14] and drug designs which affect tissue distribution [15]. There are still safety concerns for novel methods of inhaled drug development and drug aerosol administration. Safety concerns usually include pulmonary edema and underlying disease exacerbation. These safety concerns still remain to be elicited [9,11,13,16,17,18]. Several research studies have identified the parameters that influence the droplet size production, such as the following: (a) initial loading of the residual cup, (b) residual cup loading, (c) refilling of the residual cup when the initial filling has been reduced to half (this can be done only once), (d) inlet design (if used), (e) drug formulation, and (f) residual cup design [19,20,21,22]. Moreover, currently there is vast research being conducted concerning the investigation for drugs administered as aerosol for idiopathic pulmonary fibrosis (IPF) and pulmonary hypertension (PH). Firstly, for the production system, we usually use jet-nebulisers and ultra sound nebulisers. The drug composition typically consists of the following: salt, Ph, viscosity, temperature, and aerosol droplet size [4]. Finally, the underlying pulmonary disease which was thoroughly invastigated in the case of aerosol insulin and cystic fibrosis [5]. The defense mechanims of the airways and the aerosol droplet size play (<5 mass median aerodynamic size) a crucial roles in the delivery and deposition of the drug locally to the airway system [23]. One of the best methods to measure the aerosol droplets is the master sizer system, which has been previously used in several studies for aerosol medications [1,4,8]. Moreover; several modifications to the mouthpiece of the aerosol production system has been found to produce even smaller aerosol droplets [19]. The lungs have a large surface area were a drug can be rapidly absorved and with seconds enters the body. Therefore several studies are conducted to constract drugs as aerosol administration. Biological agents could be investigated as future concept for aerosol administration. In our current study we will investigate apremilast, a drug used for psoriasis whether in can be administered as aerosol after modification from tablets.

## 2. Materials and Methods

### 2.1. Drugs

The following drug was purchased: apremilast 30 mg/L tablet.

### 2.2. Nebulizers and Residual Cups

#### 2.2.1. Jet-Nebulizers and Residual Cups

Three nebulizers were chosen from our department for the experiment: Sunmist^®^ (5–7 L/min and 35 psi), Maxineb^®^ (6 L/min and 35 psi), and Invacare^®^ (4–8 L/min and 36 psi) (Figure 1).

Seven residual cups were included. Four had a capacity of ≤6 mL and three with a capacity of ≤10 mL. The large residual cups were A, D, and E (Figure 2).

The small residual cups were C, F, B and J (Figure 3). We used 1 g of powder mixed with 10 mL of water for injection for each experiment. We filled the small and large residual cups with different concentrations of the solution and performed measurements with the mastersizer.

#### 2.2.2. Ultrasound Nebulizers

The following ultrasound nebulizers were available in our department. The first was Omron^®^ NE-U07, Tokyo, Japan with a 10 mL medication cup. The second was a portable GIMA, Gessate, Italy (Choice Smart Health Care Company Limited, Wan Chai, Hong Kong, No. G2061259328002) with the following operating specifications; Particle size: 3–5 μm, Frequency: 2.5 MHz, Medication Cup Capacity: 1–6 mL. The third was a portable EASYneb^®^ II, FLAEMNUOVA, Martino, Italy with the following operating specifications; drug max capacity: 8 mL, particle size: 2.13 μm mass median aerodynamic diameter (ΜΜAD) (Figure 4).

The ultrasound nebulizers were filled with different concentrations and the measurements were performed with the mastersizer. Two factors were found to influence the MMAD response when jet nebulizers were used: drug type and residual cup design plus their interaction effect. On the other hand the MMAD for the ultrasound nebulizers was influenced by the drug concentration and ultrasound nebulizer model.

### 2.3. Measurement of Droplet Size and Droplet Size Distribution

A laser scattering apparatus (Malvern Mastersizer 2000, Malvern, Worcestershire, UK) equipped with a Scirocco dry accessory module (Malvern, Worcestershire, UK) was used for the determination of the mass median diameter of the produced particles [6,7,24,25,26].

### 2.4. Milling

The apremilast tablets were milled in a planetary ball mill (Frisch, Pulverisette-5) equipped with Agate bowls (500 mL) and 8 balls (20 mm, 20 g) with a rotational speed of approximately 800 rpm which results in an acceleration of about 7.5 g. We initiated our milling at 20 min and we acquired a mass median aerodynamic diameter (MMAD) of 3.2 μm.

## 3. Results

For jet nebulizers, the MMAD variable was transformed to a log_10_ variable since both the size frequency distribution and the boxplot information suggested so (Figure 4). The analysis of drug mean values indicated that apremilast produced mmad diameters (2.43 μm). The residual cup B contributed to greater mmad diameters as the 95% interval of mean values based on the ANOVA mean square clearly indicated, followed by cups C and F. The previous interval plot is much better clarified when the interaction means between drug and residual cap are plotted. The residual cups B, C, and F produce mmad between (2.0–3.2) (Figure 5).

For ultra sound nebulizers the mmad variable was also transformed to log_10_ base variable because values were slightly better normalized.

None main effect was observed to influence the mmad response as Table 1 clearly indicates.

Overall, none of the two groups of nebulizers seems to affect the MMAD response. However, Apremilast interacted with some residual cup designs and produced alone or jointly different MMAD diameters.

## 4. Discussion

Aerosol drugs have capabilities such as fast absorption through the alveoli. Depending on the targeted disease, patients need either fast acting or long acting drugs. A serious issue with aerosol drugs is the respiratory status of the patient. In the case of patients with severe chronic obstructive pulmonary disease (COPD) or asthma, deposition and absorption depends on status of the patients. We have different aerosol production systems for stage III and IV COPD and patients with asthma exacerbation. There is also cases wherein a patient has pulmonary infection, which again modifies the deposition and absorption of a drug. All of these factors have been extensively investigated with inhaled antibiotics and insulin [5,19]. Bronchoconstriction occurs during an exacerbation of asthma or chronic obstructive pulmonary disease (COPD) and reduces the distribution and deposition of an aerosol therapy [27,28]. However, bronchoconstriction can be observed during administration of an inhalational therapy. It is known that during infection of the airways or respiratory disease exacerbation the production of mucus also increases along with cough and reduces the absorption of an aerosol therapy since a concentration of the drug is enclosed within the mucus or is exhaled due to increased cough symptom. It is also known that there are local enzymes and transporters in the surface of the airways which differ from the larger to the smaller and interact with aerosol drugs. These different enzymes and transporters at different locations of the respiratory system modify the absorption of a drug locally [29]. Another very important factor is the mass median aerodynamic diameter <5 μm of the aerosol droplets. <5 μm is the largest size that should be administered. Another important factor for an efficient aerosol therapy is the drug design [15]. Until today jet-nebulizers have been cheaper than ultrasound nebulizers. Furthermore, several drugs are more efficiently delivered with jet-nebulizers than ultrasound nebulizers. Certainly, we should choose the aerosol production system based also on the drug that we want to deliver [22]. During our investigation of aerosol production systems it was observed that, regarding the jet-nebulizers, the residual cup design, time of nebulization, initial filling, and drug chosen affected the aerosol mist’s mean droplet size. Further reduction of the produced droplet size was achieved with the usage of inlets design in a previous study [19]. There are other parameters affecting the production or aerosol droplets for ultrasound nebulizers such as temperature (which increases from the piezoelectric crystal activity). The temperature increases when there is a higher concentration of the drug solution [30]. The temperature affects less the production of aerosol in jet-nebulizers. The combination of temperature and drug concentration causes a shift in the tension and viscosity and consequently changes the droplet size distribution (mainly for ultrasound nebulizers) [31,32]. Nebulization affects the viscosity, saturated vapor pressure, surface tension, and finally the droplet size distribution. An increase of the concentration of the aerosol droplets is mostly observed in the ultrasound nebulizers than the jet-nebulizers [33]. A variation in the viscosity is observed during the aerosol production (low the first two min and increases after the 4 min). The mean droplet size range increases when adding buffer [34]. Increased drug concentration is also a factor inducing bronchoconstriction [35]. Finally, the major factors affecting the produced aerosol mist are: (a) drug formulations (salts, buffer), (b) viscosity, (c) time of nebulization, and (d) temperature.

Apremilast acts as a selective inhibitor of the enzyme phosphodiesterase 4 (PDE4) and inhibits spontaneous production of TNF-alpha from human rheumatoid synovial cells. Apremilast is typically used against active psoriatic arthritis. [36] It is also used for the treatment of plaque psoriasis in patients who failed to respond to other systemic therapies [37]. Inhaled apremilast has advantages such as fast activation. There are publications with inhaled anti-PDE 3 and 4 agents (ensifentrine) [38,39,40,41,42]. Therefore, in this study we wanted to investigate whether apremilast could be modified from tablets to aerosol droplets with the possibility of a future clinical trial. Ensifentrine is an effective dual anti-PDE 3 and 4 agent for asthma and COPD, and therefore a second option as an anti-PDE 4 agent should be available. Indeed, we were able to create both with different jet- and ultrasound nebulisers aerosol droplets of ≤5 mmad. A future clinical trial regarding the safety of the aerosol product should be performed.

## 5. Conclusions

In this current study, we investigated drug solutions for possible future treatments. One limitation of the study was that we did not investigate the safety our drug dilution. It was observed that large concentrations >8 mL were not necessary as in our previous studies and it is only matter of how much concentration of the drug we want to deliver. Once again, the residual cup design plays an important role in the production of the aerosol droplet size and of course the drug formulation. Certainly, we can try additional local aerosol therapies for psoriasis and local administration has possible advantages. However, future experiments have to present data for a reason to use this method of administration. We should have firstly the opinion from psoriasis patients if they want another route of administration and a fast-activating drug for psoriasis.

It is possible to modify apremilast tablet to powder and produce aerosol droplets with apremilast. However, certain questions remain (namely whether the drug is safe for the lung parenchyma and whether we need fast systemic administration of this drug for treatment of psoriasis or other diseases).

## Figures and Tables

**Figure 1 ijerph-18-11590-f001:**
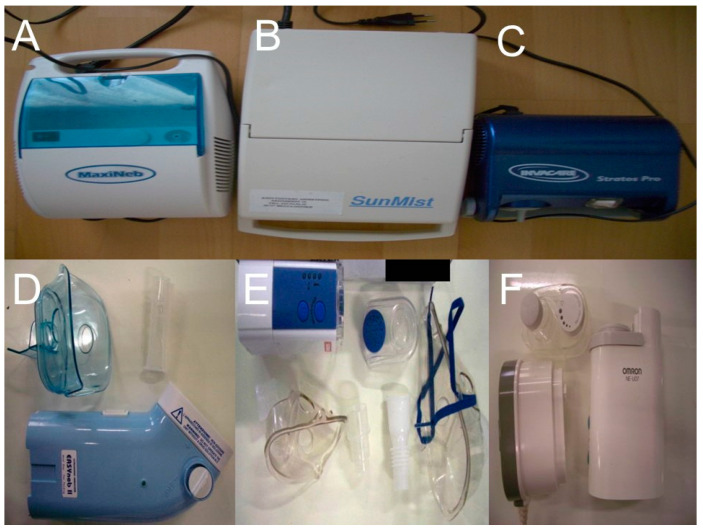
Upper row with Jet-nebulizers; (**A**) Maxineb, (**B**) Sunmist, and (**C**) Invacare. Lower row with ultrasound nebulizers; (**D**) Easyneb, (**E**) GIMA, and (**F**) Omron.

**Figure 2 ijerph-18-11590-f002:**
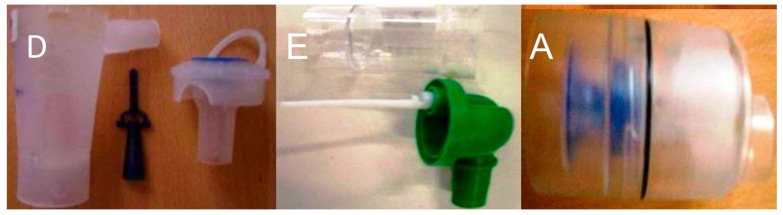
Large residual cups that can contain up to 8 mL of liquid.

**Figure 3 ijerph-18-11590-f003:**
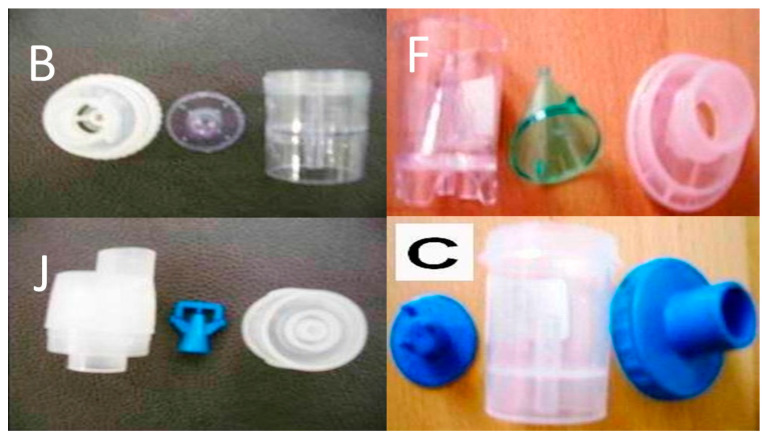
Small residual cups that can contain up to 6 mL of liquid.

**Figure 4 ijerph-18-11590-f004:**
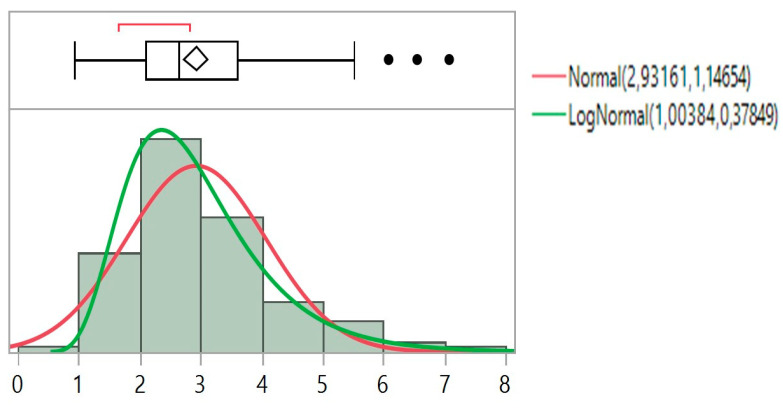
Size frequency distribution of MMAD and boxplot deployment. Dots indicate outliers and the need for a log transformation of MMAD.

**Figure 5 ijerph-18-11590-f005:**
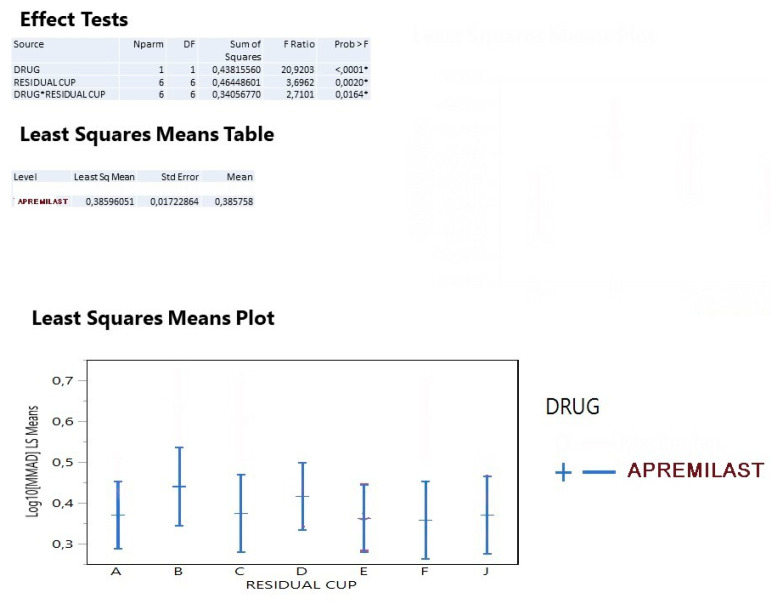
The analysis of drug mean values indicated that apremilast produced MMAD diameters (2.43 μm). The residual cup **B** contributed to greater MMAD diameters as the 95% interval of mean values based on the ANOVA mean square clearly indicated, followed by cups **C** and **F**.

**Table 1 ijerph-18-11590-t001:** AVOVA output showing the F and *p* values of the four main effects under study.

Source	Nparm	DF	Sum of Squares	F Ratio	Prob > F
DRUG	1	1	0.01397550	1.1912	0.2895
NEBULIZER	2	2	0.04570143	1.9477	0.1715
LOADING	1	1	0.01473125	1.2556	0.2772
MOUTHPIECE	1	1	0.02564812	2.1862	0.1565

## Data Availability

None to present.

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
