# Peer review of "Modification of Apremilast from Pills to Aerosol a Future Concept"

_ijerph, 2021, doi:10.3390/ijerph182111590_

Round 1
Reviewer 1 Report
There are no clear objectives or hypotheses in this study.
It is not clear about the nebulizers used and how they tested the jet nebulizers (three) with the cups (seven), and neither is the presentation of the results.
Figure 4, the x-axis is micrometers?
Better explain what each graph in Figure 5 is.
You mention that the important thing is a diameter of <5um of the aerosol droplets, which nebulizer achieved this better?
There is a lack of discussion about the results found with the nebulizers test.
Author Response
There are no clear objectives or hypotheses in this study.
Answer
Thank you for your comment
The objective was to investigate whether the apremilast can converted from i.v to aerosol and which nebulizer and residual cup is the best combination
It is not clear about the nebulizers used and how they tested the jet nebulizers (three) with the cups (seven), and neither is the presentation of the results.
Answer
Thank you for your comment
The following sentences were added in the nebulisers section
We filled the small and large residual cups with different concentrations of the solution and performed measurements with the mastersizer.
The ultrasound nebulisers were filled with different concentrations and the measurements were performed with the mastersizer.
On the other hand the MMAD for the ultrasound nebulisers was influenced by the drug concentration and ultrasound nebulizer model.
Figure 4, the x-axis is micrometers?
Answer
Yes
Better explain what each graph in Figure 5 is.
Answer
We corrected figure number 5
You mention that the important thing is a diameter of <5um of the aerosol droplets, which nebulizer achieved this better?
Answer
There was no statistical difference between the three jet-nebulisers, however; the smallset mmad was produced with the `B` residual cup
There is a lack of discussion about the results found with the nebulizers test.
Answer
We have stated in the conclusion section
`Once again the residual cup design plays an important role in the production of the aerosol droplet size and of course the drug formulation.`

Reviewer 2 Report
The paper (Modification of apremilast from pills to aerosol a future concept) under review deals with the research on the nebulisation process. The authors studied nebulization using three nebulizers. The results can be important from the point of view of pharmaceutics science. The article contains adequate and appropriately selected 42 literature items. In my opinion, the paper can be accepted for publication in the journal after corrections.
Short comments:
- In my opinion, the abstract should be corrected.
- Figs 2 and 3 are of poor quality.
- I could not find out the new academic findings in their conclusions. In my opinion, the article is interesting but requires refinement and scientific conclusions.
The above remarks do not diminish the importance of the manuscript but are intended to significantly increase its scientific value.
Author Response
The paper (Modification of apremilast from pills to aerosol a future concept) under review deals with the research on the nebulisation process. The authors studied nebulization using three nebulizers. The results can be important from the point of view of pharmaceutics science. The article contains adequate and appropriately selected 42 literature items. In my opinion, the paper can be accepted for publication in the journal after corrections.
Short comments:
- In my opinion, the abstract should be corrected.
Answer
We have added in the abstract the sentence
without any statistical difference between the different jet-nebulisers.
- Figs 2 and 3 are of poor quality.
Answer
We are sorry but we do not have any additional figures
- I could not find out the new academic findings in their conclusions. In my opinion, the article is interesting but requires refinement and scientific conclusions.
Answer
We have corrected figure number 5 and added the following sentences
We filled the small and large residual cups with different concentrations of the solution and performed measurements with the mastersizer.
The ultrasound nebulisers were filled with different concentrations and the measurements were performed with the mastersizer.
On the other hand the MMAD for the ultrasound nebulisers was influenced by the drug concentration and ultrasound nebulizer model.
The above remarks do not diminish the importance of the manuscript but are intended to significantly increase its scientific value.
Answer
Thank you for your comments

Reviewer 3 Report
Introduction part: Security aspects of aerosol drug administration should be developed deep (lines 50-52). The introduction should focus in explain why the authors have selected that drug (is there any advantadges of its administration in aerosol? (that is not discussion) How patients could benefit of it?; "a second option as an anti-PDE 4 agent should be available." L-193 - explain here in a separate paragraph).
In line 144, "Depending of the disease we need fast acting or long acting drugs." that statement should be correlationed with the drug that has been study here.
The discussion part should focus on the results found. Here is another introduction part. It is very poor. Finally, the conclusion statements are part of the material where the discussion should concentrate; limitation of the study and some observations (level of concentration ...).
Author Response
Reviewer 3
Introduction part: Security aspects of aerosol drug administration should be developed deep (lines 50-52). The introduction should focus in explain why the authors have selected that drug (is there any advantadges of its administration in aerosol? (that is not discussion) How patients could benefit of it?; "a second option as an anti-PDE 4 agent should be available." L-193 - explain here in a separate paragraph).
Answer
The following sentences have been added
Safety concerns are usually pulmonary edema and underlying disease exacerbation.
Inhaled apremilast has the ability of fast activation.
The inhaled anti-PDE 4 is not available for psoriasis so it is not of interest for this manuscript.
In line 144, "Depending of the disease we need fast acting or long acting drugs." that statement should be correlationed with the drug that has been study here.
Answer
We could use inhaled apremilast as a fast active therapy, however; do we need a fast actining drug for psoriasis. We should take this opinion from the patients first.
The discussion part should focus on the results found. Here is another introduction part. It is very poor. Finally, the conclusion statements are part of the material where the discussion should concentrate; limitation of the study and some observations (level of concentration ...).
Answer
We have added the following.
We should have firstly the opinion from psoriasis patients if they want another route of administration and a fast activing drug for psoriasis.

Round 2
Reviewer 1 Report
Accept in present form
Reviewer 3 Report
Thanks for taking into account the comments.